# Does Cold-Water Endurance Swimming Affect Pulmonary Function in Healthy Adults?

**DOI:** 10.3390/sports9010007

**Published:** 2021-01-10

**Authors:** Camilla R. Illidi, Julie Stang, Jørgen Melau, Jonny Hisdal, Trine Stensrud

**Affiliations:** 1Department of Sports Medicine, Norwegian School of Sport Sciences, 0863 Oslo, Norway; Camilla.Illidi@brunel.ac.uk (C.R.I.); julies@nih.no (J.S.); 2Prehospital Division, Vestfold Hospital Trust, 3103 Tønsberg, Norway; jorgen@melau.no; 3Department of Clinical Medicine, University of Oslo, 0315 Oslo, Norway; jonny.hisdal@medisin.uio.no; 4Department of Vascular Surgery, Oslo University Hospital, University of Oslo, 0586 Oslo, Norway

**Keywords:** cold-water swim, pulmonary function, alveolar diffusion capacity, oxygen saturation, expired nitric oxide, recreational triathletes

## Abstract

The acute effects of cold-water endurance swimming on the respiratory system have received little attention. We investigated pulmonary responses to cold-water endurance swimming in healthy recreational triathletes. Pulmonary function, alveolar diffusing capacity (*D*L_CO_), fractional exhaled nitric oxide (FE_NO_) and arterial oxygen saturation by pulse oximetry (SpO_2_) were assessed in 19 healthy adults one hour before and 2.5 h after a cold-water (mean ± SD, 10 ± 0.9 °C) swim trial (62 ± 27 min). In addition, 12 out of the 19 participants measured pulmonary function, forced vital capacity (FVC) and forced expiratory volume in one second (FEV_1_) 3, 10, 20 and 45 min post-swim by maximal expiratory flow volume loops and *D*L_CO_ by the single breath technique. FVC and FEV_1_ were significantly reduced 3 min post-swim (*p* = 0.02) (*p* = 0.04), respectively, and five of 12 participants (42%) experienced exercise-induced bronchoconstriction (EIB), defined as a ≥ 10% drop in FEV_1_. No significant changes were observed in pulmonary function 2.5 h post-swim. However, mean FE_NO_ and *D*L_CO_ were significantly reduced by 7.1% and 8.1% (*p* = 0.01) and (*p* < 0.001), respectively, 2.5 h post-swim, accompanied by a 2.5% drop (*p* < 0.001) in SpO_2_. The absolute change in *D*L_CO_ correlated significantly with the absolute decline in core temperature (*r* = 0.52; *p* = 0.02). **Conclusion:** Cold-water endurance swimming may affect the lungs in healthy recreational triathletes lasting up to 2.5 h post-swim. Some individuals appear to be more susceptible to pulmonary impairments than others, although these mechanisms need to be studied further.

## 1. Introduction

Over recent decades, aquatic endurance events such as triathlons, open-water swims, English Channel crossings and swim-runs have become more popular and are attracting participants of all ages and physical abilities [1,2,3]. However, this increase in popularity has also generated numerous reports describing acute incidences of severe dyspnea and swimming-induced pulmonary edema (SIPE) [4,5,6,7,8], impaired alveolar diffusing capacity *(D*L_CO_) and increased lung density assessed with computerized tomography [9], hypoxemia [10], hypothermia [11,12] and even death [13,14] in otherwise healthy individuals. The effects of dry-land (ultra) endurance events on pulmonary function have been thoroughly examined and reviewed [15,16], and so have the acute physiological responses to cold water immersion [17,18,19]. Still, the implications of cold-water endurance swimming on pulmonary function have received little attention.

Physiologically and mechanically, swimming is considerably different from other endurance sports: Combining exertion-induced hyperpnea and whole-body and facial immersion. Moreover, locomotor–respiratory coupling is strongly linked to stroke style and rate, only enabling the swimmer to inhale rapidly when the face is out of the water. Inspiratory muscles must therefore shorten at a high velocity against the added external resistance caused by the elevated hydrostatic pressure of the water around the thorax [20]. Thus, strenuous endurance swimming does not only induce whole-body and locomotor muscle fatigue [21], but also inspiratory muscle fatigue [22]. Furthermore, cold water temperatures pose additional risks to the immersed swimmer, including the initial “cold shock” and hyperventilation [18] and potential risk of hypothermia [11,12]. With the cold stress experienced during cold-water immersion and swimming, the respiratory system is under high constraints. Due to the popularity of (ultra) swimming events, it is therefore important to better understand the physiologic and pulmonary implications of this extreme exposure.

The effect of cold-water endurance swimming upon fractional exhaled nitric oxide (FE_NO_) is also unclear. FE_NO_ is a known measure of eosinophilic airway inflammation and is shown to be elevated in persons with asthma [23]. Similarly, increased levels of airway inflammation have been observed in swimmers and cold-air athletes (i.e., cross-country skiers and biathletes) and may be associated with repeated exposure to chlorinated environments and cold air [24]. Contrarily, however, reduced levels of FE_NO_ have been shown after exercise in cold air [25] and during immersed (head above water) exercise in cold water [26].

Based on previous research, we hypothesized that open cold-water swimming induces changes in pulmonary function, FE_NO_, *D*L_CO_ and arterial oxygen saturation (SpO_2_). Accordingly, the aim of the present study was to investigate if pulmonary function, FE_NO_, *D*L_CO_ and SpO_2_ change from before to after cold-water endurance swimming in healthy recreational triathletes.

## 2. Methods

### 2.1. Ethics Approval

The project was reviewed by the Regional Ethics Committee for Medical and Health Research (2015/1533/REK Sør-Øst), who concluded that, according to the Act of Medical and Health Research (the Health Research Act 2008), the study did not require a full review. Data handling was approved by the data protection officer at Oslo University Hospital. The study was performed according to the current revision of the Declaration of Helsinki [27], and written informed consent was obtained from all participants prior to the start of the study.

### 2.2. Participants

The present study was part of a larger cross-sectional study examining the effect of cold-water endurance swimming on core and skin temperature [13]. Nineteen healthy volunteers (Table 1) were recruited through advertisements on triathlon-related websites and social media. Inclusion criteria stated that eligible participants should be capable of swimming 3800 m continuously in ≤105 min and own a personal wetsuit in line with international triathlon competition standards [28]. Prior to inclusion, all participants completed a cardiac assessment in line with recommendations from the European Society of Cardiology [29], and reported no former medical history of pulmonary and/or cardiovascular diseases, including arrhythmias.

### 2.3. Experimental Overview

The experimental visits were performed individually by each participant over two days separated by 7–14 days. On the first day, anthropometrics, pulmonary function and SpO_2_ were assessed in the pulmonary laboratory. Immediately following the completion of the laboratory assessment, participants were transported to the seaside to start the open-water swim session within the first 60 min after the laboratory assessment was completed (see details below). Due to technical problems with the spirometer, only 12 (♂ = 7, ♀ = 5; [mean ± SD] age 34 ± 7 y; stature 177 ± 6 cm; body mass 74 ± 8 kg) of the 19 included participants measured correct pulmonary function immediately prior to the swim, and then 3, 10, 20 and 45 min post-swim. Approximately 2.5 h after the swim, the final post-swim assessments of pulmonary function, *D*L_CO_, FE_NO_ and SpO_2_ were performed in all 19 participants in the laboratory. Maximal oxygen uptake (V̇O_2max_) was determined 7–14 days after the cold-water swim.

### 2.4. Laboratory Assessment

To assess anthropometric characteristics, dual-energy X-ray absorptiometry was performed (Lunar, Prodigy Densitometer, GE Medical Systems, Maddison, WI, USA) to determine percentage body fat (BF; %) and lean body mass (LBM; kg). In stable laboratory conditions 200 m above sea level (19 ± 0.6 °C, 47 ± 5.6% relative humidity), pulmonary function was measured by forced spirometry (maximal expiratory flow–volume curves) and alveolar diffusing capacity. In line with guidelines from American Thoracic Society (ATS) and European Respiratory Society (ERS) [30], spirometry was performed in triplicate and expressed as the forced expired volume in the first second (FEV_1_), forced vital capacity (FVC) and their ratio (FEV_1_/FVC) using a spirometer (MasterScreen PFT System, CareFusion, Hochberg, Germany). For FVC and FEV_1_, the highest two reproducible maneuvers (≤150 mL) were recorded (Miller et al., 2005). Predicted values, together with age-specific lower limits of normal (LLN), were based on the reference equations from Quanjer et al. [31]. Carbon monoxide (CO) uptake in the lung, by the 8 s single-breath testing technique (MasterScreen PFT System, CareFusion, Hochberg, Germany) was measured and interpreted in line with current guidelines [32,33], and quantified as CO diffusing capacity (*D*L_CO_), CO transfer coefficient (*K*_CO_) and alveolar volume (VA).

FE_NO_ was assessed online (CLD88sp, Eco Medics, Dürtnen, Switzerland according to current guidelines from the ATS/ERS [23], with participants wearing a nose-clip. Briefly, participants inhaled to total lung capacity from ambient room air and then exhaled for 10 s, maintaining a constant expiratory flow rate of approximately 0.05 L·s^−1^ and positive mouthpiece pressure of 10–12 cmH_2_O.

Lastly, SpO_2_ was measured using a fingertip pulse oximeter (Spot Vital Signs, LXi, Welch Allyn, New York, NY, USA).

### 2.5. Field Assessment of Pulmonary Function

Immediately before the swim, 12 of the 19 participants measured pulmonary function with a portable spirometer (MasterScreen Pneumo Spirometer, CareFusion/BD, Hochberg, Germany) in line with ATS/ERS guidelines [30]. Measurements were repeated 3, 10, 20 and 45 min after the swim. Owing to equipment failure, we could only conduct field spirometry in 12 of the 19 participants.

### 2.6. Cold-Water Endurance Swim

According to the initial protocol, participants should have performed a standardized swim trial of 3800 m. However, due to a rapid and severe decline in core temperature observed in the first six participants (data reported by Melau et al.) [11], the maximum duration of the swim trial was reduced to 55 min for the remaining 13 participants, regardless of distance covered in the given time.

Water temperature was measured to 10 ± 0.9 °C; ambient air temperature was 8.0–10.0 °C and relative humidity ranged from 64–94%. As per normal race conditions, swim stroke was self-selected, and participants preferred either freestyle crawl stroke and/or breaststroke. Similarly, exercise intensity was self-selected, with most participants reporting to have completed the swim in a “controlled manner”. There were, at most, three swimmers in the water simultaneously, and no physical contact was observed between the swimmers. Immediately after exiting the water, participants were assisted into an assigned recovery room in which post-swim pulmonary function tests were performed in 12 of the 19 participants and medical surveillance was undertaken. Participants were then allowed to remove their wetsuit. Ambient conditions in the recovery room were 14–15 °C and relative humidity ranged from 64–94%.

### 2.7. Core and Skin Temperature

The assessment of skin and core temperature in the present study is described in detail by Melau et al. [11]. Briefly, skin temperature was measured with a skin sensor (YSI 400, YSI Medical Inc., Yellow Springs, OH, USA) attached to the chest, approximately 8 cm below the left clavicula. Core temperature was measured as rectal temperature (YSI 400, YSI Medical Inc., Kent, UK, inserted 10 cm past the anal sphincter. Skin and core temperature were measured (1) 15 min prior to the swim (coinciding with pre-swim pulmonary function test); (2) immediately after the swim (coinciding with first post-swim pulmonary function test); and (3) 45 min post-swim (coinciding with post-swim pulmonary function test at 45 min) [11].

### 2.8. Maximal Oxygen Uptake

VO_2max_ was assessed with a breath-by-breath gas analyzer (Oxycon Pro, Jaeger Instrument, Hochberg, Germany) on a treadmill (Bari-Mill, Woodway, WI, USA) using an incremental step protocol adapted from the Textbook of Work Physiology by Aastrand and Rodahl et al. (2003). After a 10 min individual warm-up (self-selected speed and incline of 1.7%), the treadmill was set to an incline of 5.3% and a running speed of 8–10 km·h^−1^. Running speed increased by 1 km·h^−1^ each minute until voluntary exhaustion, generally occurring at 12–14 km·h^−1^, depending on physical fitness. V̇O_2max_ was defined as the average of the two highest V̇O_2_ measurements from two 30 s intervals (Table 1) and verified by two or more of the following criteria: a V̇O_2_ plateau, a blood lactate accumulation of 8 mM, a respiratory exchange ratio of 1.15 and a heart rate of 90% of the age-predicted, maximal heart rate [34].

### 2.9. Statistical Analysis

Statistical analyses were performed using IBM SPSS Statistics (v25.0 for macOS, IBM Corp., Armonk, NY, USA). Student’s independent sample *t*-tests were performed to evaluate the differences between men and women in the descriptive and anthropometric characteristics.

Student’s paired sample *t*-tests were performed to evaluate the potential differences in pulmonary function, *D*L_CO,_ FE_NO_ and SpO_2_ before and 2.5 h after the swim. To evaluate the development in pulmonary function (i.e., FEV_1_ and FVC) from pre- to post-swim (i.e., at 3, 10, 20 and 45 min), a one-way, repeated-measures ANOVA was performed with Bonferroni corrections. Sphericity was tested in all variables using Mauchly’s test of sphericity (*p* ≥ 0.05 assumed sphericity). Pearson’s correlation coefficients were calculated to evaluate the potential relationships between changes (∆) in pulmonary function (i.e., ∆FVC and ∆FEV_1_), *D*L_CO,_ (i.e., ∆*D*L_CO_, ∆K_CO_ and ∆VA), ∆FE_NO_ and SpO_2_ and anthropometric characteristics (i.e., age, LBM and BF), change in body temperature (core and skin temperature) and swim duration. All statistical tests were performed on absolute values and changes, and the alpha level was set a priori at *p* ≤ 0.05.

## 3. Results

A total of 19 participants (seven women) completed the study. Demographic and anthropometric characteristics are presented in Table 1. Men had significantly greater body mass (*p* = 0.008) and LBM (*p* < 0.001) than women. Similarly, VO_2max_ differed between men and women (*p* = 0.01). The average swim duration was 62 ± 27 min (range 37–136 min) and did not differ between sexes (*p* = 0.455).

### 3.1. Pulmonary Function Immediately after the Swim

We observed significantly reduced FVC and FEV_1_ by 440 ± 331 mL (–8.4% ± 5.4; *p* = 0.005) and 279 ± 224 mL (–6.7 ± 4.9; *p* = 0.04), respectively, 3 min after exiting the water in 12 of the 19 participants who measured pulmonary function at the seaside. However, pulmonary function returned to baseline values within ten minutes (Figure 1, *n* = 12). Mean FEV_1_/FVC was unchanged between all timepoints (*p* = 0.33). Five participants (42%) (four men, one woman) demonstrated exercise-induced bronchoconstriction (EIB), defined as a ≥10% drop in FEV_1_ within the first 45 min post-swim. Two of the men also measured FVC and/or FEV_1_ below their age-specific LLN (range 94–99% of LLN). The average swim duration of the participants who experienced EIB was 83 ± 25 min vs. 56 ± 26 min for participants with a <10% decline. However, there were no significant relationships between swim duration and the changes seen in pulmonary function. Body fat correlated significantly with ∆FEV_1_ at 3 min post-swim (*r* = 0.79; *p* = 0.04) but not with ∆FVC at the same timepoint (*r* = −0.57; *p* = 0.39).

### 3.2. Pulmonary Responses 2.5 h Post-Swim

Pulmonary variables measured before and 2.5 h after the swim are shown in Table 2. No changes were seen in FVC and FEV_1_. However, *D*L_CO_, SpO_2_ and FE_NO_ were significantly reduced. Three male participants measured reductions in *D*L_CO_ and *K*_CO_ of 15–17.4% and 12.7–15.5%, respectively. One of these participants also demonstrated the largest drop in core temperature (from 37.7 °C to 33.35 °C; −11.6%) and skin temperature (33.7 °C to 12.9 °C; −61.7%) with SpO_2_ of 96%. The other two participants experienced mild hypoxemia, ranging from 94–95%, but with less severe body temperature changes.

Mean core temperature dropped from 36.6 ± 0.1 °C to 35.7 ± 1.1 °C (*p* < 0.001) during the swim, with an additional “after drop” of 0.6 ± 0.3 *°*C, measured at 25 min post-swim. Similarly, skin temperature dropped from 33.3 ± 0.3 °C to 19.2 ± 1.6 °C (*p* < 0.001). Additional details are shown by Melau et al. (2019).

There was a moderate correlation between the change in core temperature and change in *D*L_CO_ (*r* = 0.52; *p* = 0.02) (Figure 2). Beyond this, there were no significant correlations between changes in pulmonary function, ∆FE_NO_ or ∆SpO_2_ and the observed changes in core and/or skin temperature. Similarly, there were no significant relationships between the changes in pulmonary variables and age, sex or anthropometric indices, as well as the total swim duration.

## 4. Discussion

We investigated the pulmonary responses to a cold-water swim in healthy recreational triathletes with varying swimming skills, physical fitness and competitive level. The main findings were that mean FVC and FEV_1_ were significantly reduced immediately following the swim but returned to pre-swim levels within 10 min. However, five out of 12 (42%) experienced EIB up to 45 min post-swim. Furthermore, *D*L_CO_, SpO_2_ and FE_NO_ were significantly reduced 2.5 h after the swim and there was a moderate correlation between the change in core temperature and the change in *D*L_CO._

### 4.1. Pulmonary Function

In the present study, we found that both FVC and FEV_1_ were significantly reduced 3-min post-swim but recovered rapidly within the first 10 min post-swim. To our knowledge, there are few—if any—previous studies that have evaluated the acute effects of cold-water swimming on pulmonary function. In a series of studies, Koskela and co-workers [35,36] demonstrated how FEV_1_ and FVC may decline as a function of reduced facial skin temperature, even when inhaled air temperature is kept warm (21°C). In line with our findings, the studies also showed how FEV_1_ rapidly (≤12 min) returned to pre-exposure values after facial cooling had been terminated. The effects of facial cooling and immersion on pulmonary function have been investigated extensively, suggesting that the thermo-receptive, afferent trigeminal nerve of facial skin—stimulated by declining skin temperature and immersion—forms a vagal reflex arc causing “facial cooling-induced reflex bronchoconstriction” [35,36,37,38].

Although we did not measure facial skin temperature in the present study, we instead measured chest skin temperature. During the cold-water swim, mean chest skin temperature was almost halved (−50.2%; 33.3 ± 0.3 °C to 16.6 ± 4.3 °C) during the swim, before rapidly returning to pre-swim values when the participants had exited the water [11]. However, the absolute changes in FVC and FEV_1_ at the 3 min post-swim assessment were not correlated with the absolute changes in (chest) skin temperature, leaving room only to speculate to what extent facial cooling may have impaired pulmonary function. Yet, it is worth noting that the chest skin, to which the temperature probe was attached, was insulated from direct contact with the cold water by a waterproof, adhesive dressing and the wetsuit, and that the facial skin was in direct contact with the cold water throughout the duration of the swim trial. Thus, there is reason to believe that facial skin temperature was markedly lower than what was measured at the chest.

Bronchial hyper-responsiveness (BHR) and EIB are prevalent amongst athletes participating in endurance sports under particular environmental exposures, such as cold air and chlorine compounds in swimming pools [39,40]. In fact, evidence suggests that up to 10% of otherwise healthy individuals may experience EIB when exposed to cold air [40]. Therefore, BHR and EIB should be considered as potentially contributing factors in the individuals who were most severely affected by the swim. However, as we did not find any significant correlations between absolute change in pulmonary function and age, change in body temperature and anthropometric characteristics (with the exception of body fat percentage), we cannot clearly propose why some individuals appear to be susceptible to pulmonary function impairments following cold-water swimming. Thus, this should be considered as a potential contributing factor in the individuals who were most severely affected by the swim. The ambient air temperature during the swim ranged from 8–10 °C and the relative humidity from 64–94% (mean 81%), and so it may be questioned if this exposure is adequate to induce bronchoconstriction alone when the exercise intensity was kept low to moderate.

### 4.2. Diffusing Capacity and Arterial Desaturation

When observing reduced *D*L_CO_ following endurance exercise, pulmonary edema is brought up as a potential cause [9]. Indeed, swimming-induced pulmonary edema (SIPE) has been frequently reported in otherwise healthy adults following aquatic activities, such as triathlons [5,6,8,9], open-water swim events [4,10] and scuba diving [41,42]. The consequences of SIPE can be detrimental for the swimming human, and may include respiratory distress, hypoxemia, respiratory failure and—as suggested by Moon et al. [13]—even death.

There are several physiological factors that could make the cold-water swimmer susceptible to SIPE. For instance, mean arterial pressure increases with swimming exercise [20,43], with cold-induced peripheral vasoconstriction [17] and when wearing a tightly fitted wetsuit [44]. Together with exertion-induced hyperpnea and a restricted breathing pattern, one may exacerbate the already elevated pulmonary blood pressure to the point where mechanical failure of the pulmonary capillary wall occurs [8,45]. Moreover, the observed reduction in SpO_2_, with three participants measuring saturation levels as low as 94–95% after the swim, correlates well with the observed changes in *D*L_CO_ [46]. Nonetheless, we acknowledge that neither pulmonary auscultation, X-ray nor ultrasonography were performed to confirm the occurrence of SIPE in the present study. Interestingly, the drop in absolute values in *D*L_CO_ correlated to the drop in absolute values in core temperature. Further, VA was unchanged from pre- to post-swim, suggesting that the change in *D*L_CO_ may have been attributed to reduced vascular circulation and reduced cardiac output, as a direct consequence of hypothermia [47]. However, core temperature was normalized at the time of the post-swim laboratory assessment (2.5 h post-swim), and thus we can only speculate what may have caused the change in alveolar diffusing capacity.

### 4.3. Change in FE_NO_

The implications of cold-water endurance swimming for FE_NO_ are not fully understood. Although in warmer water temperatures (20–35 °C) than the present study, Pendergast et al. [26] demonstrated how FE_NO_ declined as a function of reduced water temperature: both at rest and during immersed (head above water) cycling exercise. At 20 °C, FE_NO_ was only 61% of that observed at 35 °C, demonstrating the measure’s sensitivity to cold ambient temperature. Similarly, Stensrud et al. [25] investigated FE_NO_ following maximal running exercise, and a subsequent 60 min recovery period, in temperate (18 °C) and cold (−10 °C) conditions in healthy individuals. Although FE_NO_ declined with a similar magnitude after exercise in both conditions (−29% in temperate vs. −33% in cold), the recovery to baseline values was significantly slower in cold conditions [25]. Our results demonstrate a less severe reduction in FE_NO_ (−7.05%) than other studies [25,26]. The timing of our post-swim assessment (2.5 h post-swim) may, however, have influenced our results. In the studies by Pendergast et al. [26] and Stensrud et al. [25], FE_NO_ was measured during and/or immediately following exercise, and one may suggest that the effect of cold-water swimming on FE_NO_ would have been more evident if measurements had been made immediately after the swim [25,26,47]. In addition, we may speculate if the type of exercise and specifically swimming may result in a lower decrease in core temperature and consequently less impact on bronchomotor nerves and NO synthase [47].

It is well established that NO synthases are temperature sensitive, and that the inhalation of cold air and/or reduced core temperature can inhibit the formation of NO from the bronchomotor nerves [47]. Moreover, NO appears to bind more readily to deoxyhemoglobin than to oxyhemoglobin [48]. Therefore, it may be hypothesized that the alveolar clearance of NO to desaturated blood might cause less lung NO to be expelled through expiration, which would also be in line with our observed drop in SpO_2_ [26,47].

As seen in our results, some participants experienced severe declines in pulmonary function after the cold-water swim. For instance, one male participant experienced absolute reductions in FVC and FEV_1_ of 1290 mL (−22.6%) and 720 mL (−15.9%), respectively, and was not fully recovered 2.5 h after the swim. Although it is unknown why some otherwise healthy individuals experience this large drop in pulmonary function, declines of a similar magnitude have been reported previously in healthy adults following facial cooling [36]. Similarly, we found that other individuals experienced severe impairments in alveolar diffusing capacity together with mild hypoxemia. These symptoms have consistently been associated with SIPE [4,9,10], but again, this could not be confirmed in the present study.

### 4.4. Methodological and Technical Considerations

The present study poses some methodological and technical considerations and limitations. The study did not include a control group performing a similar swim in temperate conditions to distinguish between the pulmonary responses to swimming as an exercise modality and those to the cold-water temperatures. In addition, the small sample size does not allow us to generalize our results to the whole triathlon population and the results must be interpreted with care.

Although none of the participants self-reported any medical history of pulmonary symptoms or asthma, we did not eliminate the presence of BHR or EIB through standardized bronchoprovocation testing [49]. Some of the more severe responses may therefore be the manifestation of undiagnosed or subclinical exercise-induced bronchoconstriction that has not been identified earlier.

We did not assess respiratory muscle contractility (e.g., maximal respiratory mouth pressures) or ventilatory capacity by maximal voluntary ventilation (MVV) to evaluate the potential influence of respiratory muscle fatigue on pulmonary function immediately after the swim. The present study showed that pulmonary function (i.e., FVC and FEV_1_) returned to pre-swim values rapidly (within 10 to 60 min). Since exercise-induced respiratory muscle fatigue usually persists for 1–2 h post-exercise, and is primarily determined by exercise intensity, it is unlikely that potential respiratory muscle fatigue significantly influenced our measures of pulmonary function.

## 5. Perspectives

In the present sport-specific study, we demonstrated that mean FVC and FEV_1_ were significantly reduced immediately following the swim but returned to pre-swim levels within 10 min. However, five out of 12 (42%) participants experienced EIB up to 20 min post-swim. Furthermore, *D*L_CO_, SpO_2_ and FE_NO_ were significantly reduced 2.5 h after the swim. Despite wearing wetsuits, healthy individuals experienced impaired pulmonary function and significant declines in arterial oxygen saturation following cold-water endurance swimming. Some individuals appear to be particularly susceptible to these impairments, but this was not associated with age, sex, anthropometry, swim duration or absolute changes in core temperature.

The mechanisms of reduced lung function and oxygen saturation after (ultra) endurance races such as triathlons and marathons may be different than after open-water endurance swimming and should thus be emphasized in further research. The practical impact of the present study is that open-water swimmers, crew and medical staff should be aware of pulmonary challenges, both the obstructive pattern observed in the present study, as well as the common symptoms of severe lung disease such as swimming-induced pulmonary edema (SIPE). We recommend swimmers with respiratory symptoms be examined for EIB before participating in endurance open-water swim races, especially in cold water.

## 6. Conclusions

This study demonstrated that healthy recreational triathletes had significantly reduced *D*L_CO_, SpO_2_ and FE_NO_ 2.5 h after cold-water endurance swimming and 42% experienced EIB and transient declines in pulmonary function 3–10 min after the cold-water swimming. Some individuals appear to be particularly susceptible to pulmonary impairments and demonstrated symptoms of SIPE. However, due to the small sample size in the present study, our findings need to be confirmed in further studies. We underline the importance of the further examination of risk factors that may result in severe pulmonary disorders during and/or after cold-water endurance swimming.

## Figures and Tables

**Figure 1 sports-09-00007-f001:**
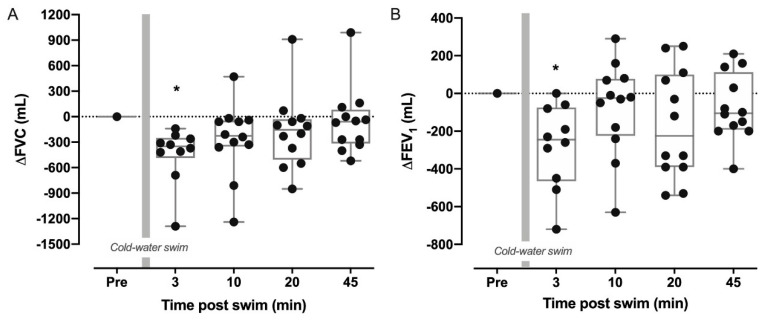
Individual responses to cold-water endurance swimming in forced vital capacity (∆FVC) (**A**) and forced expiratory volume in one second (∆FEV_1_) (**B**) at 3, 10, 20- and 45-min post swim. Boxes show median and interquartile range with whiskers of minimum to maximum range (*n* = 12). Gray bars represent cold-water swim. * Significantly different from PRE.

**Figure 2 sports-09-00007-f002:**
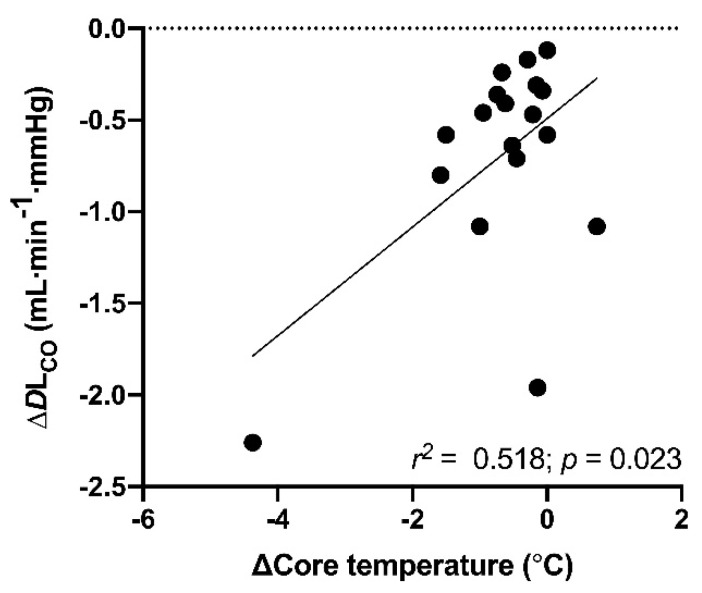
Relationship between change in core temperature and change in alveolar diffusion capacity (∆*D*L_CO_) from 1 h pre- to 2.5 h post-cold-water swim (*n* = 19).

**Table 1 sports-09-00007-t001:** Demographic, anthropometric and physiologic characteristics. Data are shown as mean ± standard deviation (SD) (*n* = 19).

	Men (*n* = 12)	Women (*n* = 7)	Total
Age, y	37.5 ± 8.4	37.1 ± 9.5	37.3 ± 8.5
Stature, cm	180.0 ± 7.0	173.8 ± 5.5	177.7 ± 7.0
Body mass, kg	83.4 ± 8.9 *	67.3 ± 7.7	77. 4 ± 13.5
Body fat, %	20.3 ± 9.4	28.5 ± 6.4	23.3 ± 9.1
Lean body mass, kg	64.6 ± 7.5 *	46.4 ± 4.9	57.9 ± 11.1
Fat mass, kg	16.1 ± 8.5	18.7 ± 5.3	17.1 ± 7.4
FVC, % of predicted	108.0 ± 15.6	107.9 ± 10.3	107.9 ± 13.5
FEV_1_, % of predicted	102.9 ± 12.8	104.1 ± 5.4	103.4 ± 10.5
FEV_1_/FVC, % of predicted	94.4 ± 6.8	95.6 ± 4.9	94.9 ± 6.0
VO_2_ max, mL∙min^−1^	5131 ± 664 *	3310 ± 474	4488 ± 1072
VO_2__max_, mL∙kg^−1^ ∙min^−1^	62.6 ± 9.8 *	49.3 ± 6.6	57.9 ± 10.8
Total training time, min∙week^−1^	588 ± 292	508 ± 243	546 ± 262
Total training time in swimming pool, min∙week^−1^	90.0 ± 77.5	99.4 ± 80.5	95.0 ± 76.4

FVC, forced expired volume; FEV_1_, forced expired volume in 1 s; V̇O_2max_, maximal oxygen uptake, * significantly different from women (*p* ≤ 0.05).

**Table 2 sports-09-00007-t002:** Pulmonary function measured before and 2.5 h post-cold-water endurance swim. Results are given as mean ± standard deviation (SD) (*n* = 19).

	Pre			Post			Change (%)
FVC, L	5.48	±	1.10	5.37	±	1.07	−2.0
FEV_1_, L	4.23	±	0.75	4.15	±	0.70	−1.9
FEV_1_/FVC, %	81.3	±	1.8	77.8	±	4.8	−4.3
*D*L_CO_, mL∙min^−1^∙mmHg	10.4	±	2.2	9.6	±	1.9	−7.6 *
VA, L	6.51	±	1.11	6.39	±	1.04	−1.8
*K*_CO_, mL∙min^−1^∙mmHg∙L^−1^	1.59	±	0.17	1.50	±	0.19	−5.7 *
SpO_2_, %	98.7	±	1.1	96.2	±	1.1	−2.5 *
FE_NO_, ppb	20.2	±	8.8	18.1	±	7.7	−7.05 *

FVC, forced vital capacity; FEV_1_, forced expired volume in 1 s; DL_CO_, diffusing capacity for carbon monoxide; VA, alveolar volume; K_CO_, transfer coefficient for CO; SpO_2_, arterial oxygen saturation; FE_NO_, fractional exhaled nitric oxide. * Significant difference between pre- and 2.5 h post-swim assessment (*p* ≤ 0.05).

## Data Availability

The data presented in this study are available on request from the corresponding author. The data are not publicly available due to privacy agreement between participants, principal investigator and Oslo University Hospital, as per participant consent form.

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
