# Peer review of "Does Cold-Water Endurance Swimming Affect Pulmonary Function in Healthy Adults?"

_sports, 2021, doi:10.3390/sports9010007_

Round 1

Reviewer 1 Report

This study analysed the effects of cold-water endurance swimming in healthy individuals on pulmonary function. Overall this is a very interesting study, and as the authors acknowledge in the introduction, the topic has been under-investigated in the past despite the increase in popularity of these sports among non-professional athletes. 

Despite the novelty, I have some concerns about the methodology. As the authors point out, the major problem with the study is that there's no control group comparing how pulmonary function might respond after an endurance swimming in thermo-neutral water. If this was not possible during the investigation, I suggest the authors to look at previous research to investigate whether such phenomenon occurs under these conditions. In the absence of a control group, this might shed some light to the implications of the study. Maybe PFT decrease immediately after swimming given the strain on the inspiratory and expiratory muscles or as a result of water immersion and the effects of hydrostatic pressure over the thorax (which I also suggest should be incorporated in the discussion). 

The other major concern is the limited number of participants who participated in the study. It might be too bold to draw conclusions on the changes in pulmonary function with only 12 people being investigated. In addition, based on the fact that it was not possible to determine whether any unknown underlying respiratory condition might have influenced the pulmonary responses, conclusions of this study should be more discrete. I suggest instead that the authors use this study as proof of principle on how swimming in cold water might negatively affect pulmonary function. 

Finally, i think the discussion should incorporate more information on the hypothesis of why swimming in cold water might have caused a decrease in pulmonary function. In my opinion this part of the discussion is missing on potential explanations for these while these are provided for other variables such as FEno and diffussing capacity. 

Other minor corrections are that after two points (:) no capital letter should follow (see for instance line 59) and that the sentence between lines 250 and 252 seems incomplete somehow or it's not well constructed. 

Author Response

We would like to thank the reviewer for providing comprehensive and informative feedback on the manuscript. Please find below their specific comments along with our annotated responses. We have stated the line numbers of any amendments, and the inline manuscript revisions are highlighted in yellow for ease of reference.

RESPONSE TO REVIEWER 1

As the authors point out, the major problem with the study is that there's no control group comparing how pulmonary function might respond after an endurance swimming in thermo-neutral water. If this was not possible during the investigation, I suggest the authors to look at previous research to investigate whether such phenomenon occurs under these conditions. In the absence of a control group, this might shed some light to the implications of the study. Maybe PFT decrease immediately after swimming given the strain on the inspiratory and expiratory muscles or as a result of water immersion and the effects of hydrostatic pressure over the thorax (which I also suggest should be incorporated in the discussion). 

We thank the reviewer for the pertinent comment. Naturally, we agree with the notion that there should, ideally, have been a control group completing an endurance swim in thermo-neutral water. However, this was not possible due to time constrains and the added complexity of the study design.

To our knowledge, there are few – if any – former studies investigating the influence of endurance swimming in thermo-neutral water on pulmonary function. However, there are a number of studies (e.g., Hill et al., Med Sci Sport Ex, 1991, 23(11),1260-1264), investigating pulmonary function (i.e., spirometry indices) following Ironman triathlons; in which swimming was performed in thermo-neutral water (22°C). However, because pulmonary function was only assessed pre- and post-triathlon; thus, including also 180 km cycling and 42.2 km running, it was not considered appropriate to compare these studies with ours. We have, to the best of our capability, utilised and compared our findings with the most appropriate references available at the time of writing.

Regarding the suggestion about the potential influence of respiratory muscle fatigue on measures of pulmonary function, we have now incorporated a paragraph about this in the methodological considerations (lines 335-41).

The other major concern is the limited number of participants who participated in the study. It might be too bold to draw conclusions on the changes in pulmonary function with only 12 people being investigated. In addition, based on the fact that it was not possible to determine whether any unknown underlying respiratory condition might have influenced the pulmonary responses, conclusions of this study should be more discrete. I suggest instead that the authors use this study as proof of principle on how swimming in cold water might negatively affect pulmonary function. 

After a full revision, we hope that the reviewer will find the manuscript approved. For instance, we have moderated our conclusions and points of discussion to be more inquisitive and discrete rather than resolute. Nineteen participants are included in the present study and they measured lung function 1h before and 2.5h after the swim, however only 12 of them measured lung function immediately before and after the swim at the swim site due to problems with the spirometer. We have tried to clarify this in the methods, lines84-92 ;115-118; 132 and 179-83. Moreover, we have pointed out where more research is warranted, as to indicate that our study is a ‘proof of principle’ (e.g. lines 328-29; 361-67).

Finally, i think the discussion should incorporate more information on the hypothesis of why swimming in cold water might have caused a decrease in pulmonary function. In my opinion this part of the discussion is missing on potential explanations for these while these are provided for other variables such as FEno and diffussing capacity. 

Thank you for that comment, we have completed a full revision of the manuscript, and we hope that the reviewer will find the discussion improved. The changes are marked in yellow on lines 245-52;256-62.

Other minor corrections are that after two points (:) no capital letter should follow (see for instance line 59) and that the sentence between lines 250 and 252 seems incomplete somehow or it's not well constructed. 

Thank you for that remark, this is now corrected

Reviewer 2 Report

General comments:  Illidi et al investigated pulmonary function both pre-exercise and 2.5 hours post-exercise (exercise consisted of a 37-136 minute open water endurance swim in cold water, with ambient temperatures between 8-10ºC) in 19 subjects. Twelve of these subjects also underwent pulmonary function testing immediately prior, then 3-, 10-, 20-, and 45-minutes post-swim, as part of a larger study investigating serial body temperature changes. Significant pre- to post-swim decreases were found in lung diffusing capacity (DLco) arterial oxygen saturation (SpO2%), and fractional exhaled nitric oxide (FeNO) for the 19 swimmers tested. Additionally, 33% of 12 athletes serially tested developed exercise-induced bronchoconstriction (EIB) up to 20-minutes post-swim. Practical recommendations caution athletes and medical staff to be cognizant of the potential for respiratory challenges during and after cold-water swimming, particularly those with a history of pulmonary dysfunction.

Although limited in subject numbers (and lack of a full-data set for serial testing by the swimming venue) and stated technical challenges, these data do provide a useful window into “real-life” physiology under the added stress of extreme cold. Thus, with a few suggestions I believe that these data represent useful pilot data to launch further investigations evaluating pulmonary function (adaptations versus disease) in long-distance swimmers. These data are particularly relevant to future strategies aimed at reducing deaths in future cold-water endurance swim events.

Specific comments:

There is confusion regarding data presentation from the two separate data sets (N=19 and N=12). These data would actually be more scientifically manageable if the authors stuck with the full dataset cohort (N=12), but there is likely a reason to present both data sets. Thus, my questions are: 1) are there statistical differences between the N=19 versus N=12 and 2) Do the statistical differences change (i.e. statistical significance in the N=19 cohort but not in the N=12 cohort, pre- to 2.5 hour post-measurement?). These need to be addressed and defended, particularly if the authors wish to present both data sets (and not streamline down to the N=12 full data).

Introduction Section:

This is a picky point, but the authors switch between British spellings (i.e. dyspnea line 30; hyperpnoea, line 38) and American English spellings (hypoxemia, line 32). Please pick one or the other (the American English spellings dominate) in the final proofreading stage (and identify other typographical errors, like FENO versus FENO).

Lines 59-61: the authors do not critically address (or present) the influence of age, sex, or body composition other than at baseline as demographic data. Plus, the cohort is too small to address meaningful comparisons of changes due to age, sex, or body composition. I would recommend removing this sentence, as a main study aim (especially since there were no significant correlations and data not presented).

Methods section:

Line 72: I believe that reference 13 should be replaced by reference 11

Lines 78-80: Please provide more detailed explanation on when subjects were tested over the two days? Specifically, how long was the time period between baseline lab testing before the start of the swim? 2 hours? 24 hours?? What was the ambient temperature in the lab and was it consistent?

Line 116: please provide an actual reference after the author (i.e. reference 11), to be consistent.

Results section:

This is where the data get most confusing. Please try and better organize these data into two separate data sets, if the authors choose to include both the N=19 and N=12 analyses. For example, how many females were in the N=12 cohort? Where there statistically significant differences in the baseline (pre) and 2.5 hour (post) values between the cohorts (including swim time)?

Lines 195-197: Since core temperature was measured 15-minutes prior up to 45-minutes into recovery (Lines 131-133), what data points were actually used (i.e. from what range)? This timeframe is inconsistent with baseline versus 2.5-hours post and warrants further clarity. Can you show this relationship (Figure), as it seems physiologically important.

Discussion section:

The one participant’s core temperature drop to 33.5ºC is profoundly (clinically) concerning – do the authors believe this is a true reading?? Did this subject experience need medical care for severe hypothermia??

Just curious, since your water and ambient temperatures were very low, is there literature to support increased mortality with decreasing water and/or ambient temperature??

Author Response

We would like to thank the reviewer for providing comprehensive and informative feedback on the manuscript. Please find below their specific comments along with our annotated responses. We have stated the line numbers of any amendments, and the inline manuscript revisions are highlighted in yellow for ease of reference.

RESPONSE TO REVIEWER 2

Specific comments:

There is confusion regarding data presentation from the two separate data sets (N=19 and N=12). These data would actually be more scientifically manageable if the authors stuck with the full dataset cohort (N=12), but there is likely a reason to present both data sets. Thus, my questions are: 1) are there statistical differences between the N=19 versus N=12 and 2) Do the statistical differences change (i.e. statistical significance in the N=19 cohort but not in the N=12 cohort, pre- to 2.5 hour post-measurement?). These need to be addressed and defended, particularly if the authors wish to present both data sets (and not streamline down to the N=12 full data).

We acknowledge this very appropriate point and apologise for any confusion. We have included only one study population, including 19 participants. However, due to technical problems with the spirometer only 12 out of the 19 participants measured lung function immediately before and after the swim. We have now tried to clarify this in the method section lines84-92 ;115 and 130-34. We have also explained this in the result section and hope the reviewer will find the methods and results clearer (see lines 179-83). By clarifying the study population, we suggest that the statistical analyses the reviewer ask for is not necessary. We have, however, described the 12 participants in lines 86-88.

Introduction Section:

This is a picky point, but the authors switch between British spellings (i.e. dyspnea line 30; hyperpnoea, line 38) and American English spellings (hypoxemia, line 32). Please pick one or the other (the American English spellings dominate) in the final proofreading stage (and identify other typographical errors, like FENO versus FENO).

Thank you for pointing this out. The suggested changes have been changed, and the manuscript now holds American English standard, as suggested. We have also corrected typographical errors.

Lines 59-61: the authors do not critically address (or present) the influence of age, sex, or body composition other than at baseline as demographic data. Plus, the cohort is too small to address meaningful comparisons of changes due to age, sex, or body composition. I would recommend removing this sentence, as a main study aim (especially since there were no significant correlations and data not presented).

Thank you for these very appropriate comments. We agree that the study population is too small and have now removed the influence of age, sex and body composition as part of the main aim.

Method section

Line 72: I believe that reference 13 should be replaced by reference 11

Lines 78-80: Please provide more detailed explanation on when subjects were tested over the two days? Specifically, how long was the time period between baseline lab testing before the start of the swim? 2 hours? 24 hours?? What was the ambient temperature in the lab and was it consistent?

Line 116: please provide an actual reference after the author (i.e. reference 11), to be consistent.

Thanks for that comments, we have now replaced ref 13 with ref. 11 and included the requested details in the methods section (lines 82-92 and 96-97). In addition, ref. 11 is included after the author in line 122.

Result section

This is where the data get most confusing. Please try and better organize these data into two separate data sets, if the authors choose to include both the N=19 and N=12 analyses. For example, how many females were in the N=12 cohort? Where there statistically significant differences in the baseline (pre) and 2.5 hour (post) values between the cohorts (including swim time)?

Thank you again for these very appropriate questions and comments. We apologise for any confusion. As we have now clarified the study population in the methods, we hope that we have addressed your concerns and questions regarding the sample size and study population.

Lines 195-197: Since core temperature was measured 15-minutes prior up to 45-minutes into recovery (Lines 131-133), what data points were actually used (i.e. from what range)? This timeframe is inconsistent with baseline versus 2.5-hours post and warrants further clarity. Can you show this relationship (Figure), as it seems physiologically important.

Thank you, these concerns have now been addressed in the manuscript (lines 140-43).

Discussion section

The one participant’s core temperature drop to 33.5ºC is profoundly (clinically) concerning – do the authors believe this is a true reading?? Did this subject experience need medical care for severe hypothermia??

Just curious, since your water and ambient temperatures were very low, is there literature to support increased mortality with decreasing water and/or ambient temperature??

We agree, a drop in core temperature of this magnitude is concerning. This particular participant exhibited normal core temperature at pre-swim assessment (37.7°C), and there was no reason to believe that the measurement tool (rectal probe; YSI 400, YSI Medical Inc., Yellow Springs, US) was damaged/malfunctioning based on post-hoc evaluation of the data and the participant’s vital signs upon exiting the water. The rectal temperature probes utilised in this particular study have reported accuracy of ± 0.1 C from 32 to 42 C. Based on these considerations, we are certain that this is a true reading.

This participant was, like the remaining participants, under surveillance of on-site medical personnel for 60 minutes post-swim. The participant responded rapidly to thermo-neutral conditions and experienced no medical difficulties during the recovery from this trial.

Unfortunately, we do not know any research supporting increased mortality with decreasing water and/or ambient temperature.

Reviewer 3 Report

General comments

The abstract is clearly written and establish nicely the state of the art allowing to point out the gaps in the current knowledge of this activity. However, in order to avoid some misinterpretation at the first read, I suggest the authors to precise when the studies mentioned in the text focuses on land activities (see my specific comments). Furthermore, the authors should formulate some hypothesis to strengthen their rational and experimental protocol.

The protocol and the findings from the present study are of particular interest for open-water events training and safety purposes. Unfortunately the methods and discussion sections suffer from numerous mistakes or lack of precision that should be reconsider for publication of the manuscript. I detailed below several comments to consider in order to precise some key points that, I hope, could facilitate readers comprehension and provide further discussion details.

Specific comments:

L 33 : The authors should precise whether the ultra endurance activities referred here to land (or dry) activities to facilitate understanding of the current knowledge.

L 49 : Please indicate for the first occurrence in the introduction section what FENO refers to.

L 75 : The link provides to present the international standards regarding the wetsuit is no longer available. Due to the different thinness of the neoprene corresponding to the international standards, could the authors precise whether all athletes wore similar wetsuits or whether some important differences were noted between athletes, which could likely influence the level of neuromuscular fatigue of the locomotor muscles, and change in core temperature during swimming.

L 80 : Were the two days of experimentation consecutives or separated?

L 83-86 : please precise how was managed the 2.5 hours resting period for the participants involving in the different pulmonary function assessment? Were they allow to remove their wetsuits? What was the temperature of the laboratory the participants stayed in? These additional information will provide to better consider the changes occurring during this period.

L 86-87 : performing VO2MAX assessment 7 to 14 days after the experimental session is somewhat unusual and would let me think that this measure was added to an original protocol. Did the authors ensured that participants did not contracted – and were free of - any illness following the experimental session that could possibly impaired their performance level?

L 103-104 : did the authors ensured that the position of the trunk remained stable during the test to alleviate any confounding effect of pressure effects on pulmonary testings?

L 106-107 : The prolonged immersion of the hand in the cold water would likely have induced a vasoconstriction effect resulting in finger hypo perfusion that could alter SpO2 measurement (e.g. Nishiyama T., 2006, Can J Anaesth, 53(2):136-8). Did the authors checked for skin temperature of the finger, and not only that of the chest, to perform reliable measurements?

L 117 : Please indicate the duration or the range of durations of the event for the remaining participants (≤55 min being not enough acurate).

L 134 : The description of the incremental treadmill protocol used in the present study lacks of clarity. First, did the participants performed a warm-up before the incremental protocol? Even for trained athletes, starting an incremental testing at 8 or 10 km/h with a 5.3% slope appear very demanding and would impact the proper evaluation of VO2MAX. Then, please precise whether starting speed was set at 8 or at 10 km/h, since under the present form (i.e. 8-10 km/h), the protocol let me think that the speed was different between the participants. Could you also please report the mean velocity associated to VO2MAX in order to have an idea of the performance of the athletes and the duration of the test? In addition, you mentioned that two values were recorded to define VO2MAX, but please specify whether the values presented in table 1 are the average of the two measured values or I only the greater value was considered?

L 146-148: Could the authors justify why they used two different analyses (i.e. student’s or ANOVA) for the pulmonary function, FENO and SpO2 vs FEV1 and FVC? Did the authors checked for the normality and sphericity of the variables to apply the ANOVA?

L 196-197: Please report the values for r and p of these relationships.

L 210-213: In accordance with one of my previous comment in the method section, it could be of interest to indicate the temperature of the resting room to compare the present findings with those of Koskela and colleagues mentioned here.

L 219-221: given the impressive drop in chest temperature during swimming and after exiting, was the temperature of the skin, or rather the temperature at the surface of the skin that was measured? Indeed, the position of the probe at the surface of the skin would rather measure water temperature in the wetsuit, therefore, did you ensure that the probe was isolated from water?

L 242-243: you mentioned previously that the participants reported that they performed the swimming exercise in a controlled manner. In this context, could the authors precise what they refer to “controlled manner” since it could be suggested that this intensity would not induced post-exercise hyperpnoea, or of a low to moderate amplitude, and would therefore moderately contribute to SIPE. Or maybe the authors should reconsider this mechanism to explain SIPE.

L 260-261: Additionally to the difference in time-delay measurement between the present and previous mentioned studies, the type of exercise, and particularly the involvement of the whole body during exercise in the present study could also represent a factor to consider for the different in FENO drops. Indeed, given the involvement of the whole body during swimming compared to cycling, one could suggest at lower decrease in core temperature and consequently a lesser impact on bronchomotor nerves and NO synthase. Authors should discuss this hypothesis if this could represent a cofounding factor for this variable.

Author Response

We would like to thank the reviewer for providing comprehensive and informative feedback on the manuscript. Please find below their specific comments along with our annotated responses. We have stated the line numbers of any amendments, and the inline manuscript revisions are highlighted in yellow for ease of reference.

RESPONSE TO REVIEWER 3

General comments

The abstract is clearly written and establish nicely the state of the art allowing to point out the gaps in the current knowledge of this activity. However, in order to avoid some misinterpretation at the first read, I suggest the authors to precise when the studies mentioned in the text focuses on land activities (see my specific comments). Furthermore, the authors should formulate some hypothesis to strengthen their rational and experimental protocol.

The protocol and the findings from the present study are of particular interest for open-water events training and safety purposes. Unfortunately the methods and discussion sections suffer from numerous mistakes or lack of precision that should be reconsider for publication of the manuscript. I detailed below several comments to consider in order to precise some key points that, I hope, could facilitate readers comprehension and provide further discussion details.

Thank you for your important comments, we have now specified when the referred studies focus on land-based activities (see specific comments below). We have also included a hypothesis (line 60-61) in the introduction for strengthening our experimental protocol and rationale.

Thank you for your thorough review of the manuscript, we have now tried to improve the method and discussion section and answered your specific comments point by point.

Specific comments:

L 33 : The authors should precise whether the ultra endurance activities referred here to land (or dry) activities to facilitate understanding of the current knowledge.

Thank you, this has now been specified in the introduction (line 36). Hopefully, we have made it clearer to the reader if comparative studies are performed on dry-land or in water; the mode of exercise and ambient temperatures in the cited studies in the discussion chapter. We hope that this may help the reader contextualize our study and findings in relation to previous research. 

L 49 : Please indicate for the first occurrence in the introduction section what FENO refers to.

Thank you for that remark, this has now been specified in upon first occurrence in the introduction (line 52).

L 75 : The link provides to present the international standards regarding the wetsuit is no longer available. Due to the different thinness of the neoprene corresponding to the international standards, could the authors precise whether all athletes wore similar wetsuits or whether some important differences were noted between athletes, which could likely influence the level of neuromuscular fatigue of the locomotor muscles, and change in core temperature during swimming.

We have now updated the attached link to the reference [28]. Participants were asked to wear their personal swim-specific wetsuit (i.e., not a diving or surfing wetsuit) which they would normally use for their triathlon races. Although the wetsuits were of different brands (e.g. 2XU, Huub, Orca), all participants wore newer models without impairments due to excessive wear and/or reduced quality. Although the use of wetsuit will likely restrict limb range of motion (e.g., Nessler et al., Plos ONE; 2015, 10(11):e0142325), swim-specific wetsuit models are specifically designed (for instance, see https://www.2xu.com/au/wetsuits.html?lang=en) to elicit free and unrestricted range of motion of shoulders and elbows, thus reducing the severity of neuromuscular fatigue of upper body muscles (e.g., deltoids). Furthermore, a tightly-fitted wetsuit, without excessive wear, will have little-to-no waterflow through the wetsuit, and thus protect the swimmer by acting as an insulator from the cold water. Based on these considerations, we are confident that the wetsuits worn by our participants did not elicit excessive neuromuscular fatigue (beyond what could have been expected during a swim in thermo-neutral water) and that the wetsuit only served as an insulator for the protection of core temperature

L 80 : Were the two days of experimentation consecutives or separated?

Thank you, this is now pointed out in the manuscript (lines 83 and 92). The two days were separated by 7-14 days.

L 83-86 : please precise how was managed the 2.5 hours resting period for the participants involving in the different pulmonary function assessment? Were they allow to remove their wetsuits? What was the temperature of the laboratory the participants stayed in? These additional information will provide to better consider the changes occurring during this period.

Thank you for this very appropriate comment. We have now specified the ambient conditions in 1) the laboratory at pre-post swim assessment (line 96-97); 2) during the field assessment by the seaside in the 2.5 h resting period following the swim (lines 130-34) .We have also provided details about the use of wetsuit (e.g., when the wetsuit was worn, when it was removed, and which tests were performed with wetsuit) (lines 130-34).

L 86-87 : performing VO2MAX assessment 7 to 14 days after the experimental session is somewhat unusual and would let me think that this measure was added to an original protocol.

Thank you for this appropriate question. We agree that 7-14 days from the experimental swim to the assessment of VO2max was unusual. However, this delay in data collection was due to multiple reasons. First, laboratory availability was limited when the study was conducted, and it was therefore difficult to book the laboratory for testing of 19 participants within a short period of time (e.g., three days) for the second visit. Secondly, most participants were working full-time jobs, and were therefore restricted in their time availability for testing. We therefore saw no other option than to invite participants to the laboratory for testing over the course of 14 days.

L 103-104: Did the authors ensured that participants did not contracted – and were free of - any illness following the experimental session that could possibly impaired their performance level?

As with most research studies with humans, during which participants visit the laboratory on multiple occasions, we could not guarantee that participants did not contract an illness following the first experimental session. However, following the swim-trial, participants underwent a medical surveillance for ≥ 60 min during which time their wetsuit was removed (within the first five minutes post-swim); they were provided with towels to dry properly and were given (emergency) foil blankets to normalize core- and skin temperature rapidly. Furthermore, prior to VO2max testing, participants were requested to re-schedule their test if they exhibited any symptoms of illness, and filled out a standardized health questionnaire prior to their test, ensuring that they were in a healthy state.

L 103-104 : did the authors ensured that the position of the trunk remained stable during the test to alleviate any confounding effect of pressure effects on pulmonary testings?

All participants performed pulmonary function tests according to the established guidelines provided by the European Respiratory Society/American Thoracic Society (e.g., Miller et al; Eur Respir J, 2005, 26, 319-338) by experienced test leaders. This also included the use of appropriate, standardized equipment which was calibrated according to the established guidelines. We are, therefore, confident that there was little to no confounding effect of instable trunk, and a consequent increase in elastic and/or resistive pressure, on measures of pulmonary function.

L 106-107 : The prolonged immersion of the hand in the cold water would likely have induced a vasoconstriction effect resulting in finger hypo perfusion that could alter SpO2 measurement (e.g. Nishiyama T., 2006, Can J Anaesth, 53(2):136-8). Did the authors checked for skin temperature of the finger, and not only that of the chest, to perform reliable measurements?

Thank you for raising this point, and we fully agree with the reviewer that cold-induced vaso-constriction of the finger may cause hypoperfusion, potentially influencing SpO2 measurements. Although we did not validate our pulse-oximeter with an additional pulse-oximeter, or checked the skin temperature of the finger, our temperature data showed that the majority of participants measured pre-swim [chest] skin temperature within the first 45 min post-swim, and that core temperature was normalized within the first 60 min post-swim (Melau et al., Sports, 2019, (7)130, 1-9). Seeing that post-swim assessment of SpO2 was performed 2.5 h post-swim, we therefore do not think it is likely that cold-induced vasoconstriction of arms and fingers influenced our measures of SpO2.

L 117 : Please indicate the duration or the range of durations of the event for the remaining participants (≤55 min being not enough acurate).

Thank you for that remark, this is now specified in the given section (line 120-24).

L 134 : The description of the incremental treadmill protocol used in the present study lacks of clarity. First, did the participants performed a warm-up before the incremental protocol? Even for trained athletes, starting an incremental testing at 8 or 10 km/h with a 5.3% slope appear very demanding and would impact the proper evaluation of VO2MAX. Then, please precise whether starting speed was set at 8 or at 10 km/h, since under the present form (i.e. 8-10 km/h), the protocol let me think that the speed was different between the participants. Could you also please report the mean velocity associated to VO2MAX in order to have an idea of the performance of the athletes and the duration of the test? In addition, you mentioned that two values were recorded to define VO2MAX, but please specify whether the values presented in table 1 are the average of the two measured values or I only the greater value was considered?

Thank you for these pertinent questions; we have addressed the reviewer’s questions in the specific section in the methods.

1)    Yes, participants performed a standardized warm-up of 10 min. This has now beeing outlined in the methods section (line 147-48).

2)    The starting speed was indeed different between participants, and ranged from 8-10 km/h, depending on prior running performance and experience of the participant. The applied protocol has been adopted from the works of Aastrand et al (in: Textbook of Work Physiology: physiological bases of exercise; 2003, by Dahl et al.). In general, participants increased running speed 4-5 times, thus terminating the test at 12-14 km/h, depending on their level of fitness. It must furthermore be noted that all participants reached at least two or more of the criteria for VO2max (incl. plateau in VO2).

3)    This has now been specified in the methods section (lines 147-53).

4)    This has now been specified in the methods section. (line 150-51).

L 146-148: Could the authors justify why they used two different analyses (i.e. student’s or ANOVA) for the pulmonary function, FENO and SpO2 vs FEV1 and FVC? Did the authors checked for the normality and sphericity of the variables to apply the ANOVA?

Paired samples student’s t tests were used to assess the statistical difference in pulmonary function from pre- to post-swim (laboratory measurements; i.e., FENO, SpO2, FEV1, FVC, DLCO, KCO and VA) for the whole study population (n=19). On the other hand, one-way, repeated-measures ANOVA was used to assess the statistical differences in pulmonary function immediately before and after the swim (field measurements; i.e., FEV1, FVC, FEV1/FVC) (n=12)

We treated laboratory-assessed spirometry parameters (i.e., FVC, FEV1 and FEV1/FVC) differently from those measured during field assessment due to multiple test points at seaside and differences in measurement equipment (MasterScreen PFT System, CareFusion vs. MasterScreen Pneumo Spirometer, CareFusion). However, both spirometers were calibrated according to ambient conditions and altitude in the laboratory and at the seaside.

Prior to performing parametric statistics (i.e., t tests and ANOVA), normal distribution was checked using Shapiro-Wilk test, and sphericity was checked using Mauchly’s Test of Sphericity, and we concluded that all variables met the criteria for parametric statistics. The statistics section is now edited, and we hope the reviewer will find it improved (lines 161-67).

L 196-197: Please report the values for r and p of these relationships.

Thank you for this very appropriate request. This has now been included in the result section (lines 188-89 and 215-17). To further enhance clarity in this section, we have included a correlation plot showing the significant correlation between change in core temperature and change in DLCO (Figure 2).

L 210-213: In accordance with one of my previous comment in the method section, it could be of interest to indicate the temperature of the resting room to compare the present findings with those of Koskela and colleagues mentioned here.

This has now been specified in the methods section (lines 133-34).

L 219-221: given the impressive drop in chest temperature during swimming and after exiting, was the temperature of the skin, or rather the temperature at the surface of the skin that was measured? Indeed, the position of the probe at the surface of the skin would rather measure water temperature in the wetsuit, therefore, did you ensure that the probe was isolated from water?

The reviewer raises some very pertinent questions. In the present study, we used YSI 40 skin temperature probes (YSI 409, YSI Medical Inc., Yellow Springs, US). The probes were attached with waterproof adhesive dressing three centimeters above the nipple. In addition to the waterproof adhesive dressing, the compression posed by the tightly fitted wetsuit worn by participants is supposed to limit waterflow inside the wetsuit. Thus, we are confident that the skin temperature probe was adequately insulated from surrounding cold water and emphasize that the probe was never in direct contact with cold water; only skin.

We also wish to add that similar declines in skin temperature have been shown in previous studies of similar design (e.g., Rüst et al; Extrem Physiol Med, 2012, 1(8); doi: 10.1186/2046-7648-1-8 and Tipton et al; J Appl Physiol, 1991,70(1) 317-322; doi: 10.1152/jappl.1991.70.1.317). Thus, the observed declines in skin temperatures were indeed impressive, although not unwarranted. In lines 245-52, we have now discussed these questions in relations to the absolute change in FEV1, and the potential influence of facial-skin cooling on bronchoconstriction.

L 242-243: you mentioned previously that the participants reported that they performed the swimming exercise in a controlled manner. In this context, could the authors precise what they refer to “controlled manner” since it could be suggested that this intensity would not induced post-exercise hyperpnoea, or of a low to moderate amplitude, and would therefore moderately contribute to SIPE. Or maybe the authors should reconsider this mechanism to explain SIPE.

Thank you for that comment, the meaning of a “controlled manner” in our study is that the participants decided the exercise intensity by themselves depending on aerobic capacity and length of the swim. They were all experienced recreational triathletes and knew which intensity level they were able to keep throughout the swim.

Our research group found also reduced lung function, FEV1 and FVC in participants after Norseman Triathlon and we suggest that other factors than high intensity may induce EIB, for example respiratory muscle fatigue after long distance endurance performance at low intensity and also exercise in unfavorable environmental conditions (Stensrud et al, 2020).

Adir et al 2004 found that recreational swimmers experienced SIPE after long distance open water swim. The diagnosis is challenging, and the most reporting symptoms and findings are dyspnea and/or chest tightness, hemoptysis, reduced lung volumes, reduced oxygen saturation and diffusion capacity. In addition, radiological findings compatible with pulmonary edema. Since our aim in the present study was to investigate pulmonary function, alveolar diffusing capacity and FENO, as well as SpO2 after cold-water endurance swimming, we have only described symptoms that can explain SIPE in three of the participants and this is not confirmed by radiological examination. However, Grünig et al 2017 conclude in a review that SIPE can be diagnosed based on history, and clinical presentation. Diagnostic tests including monitoring of oxygen saturation and that a radiological examination could support the diagnosis.

L 260-261: Additionally to the difference in time-delay measurement between the present and previous mentioned studies, the type of exercise, and particularly the involvement of the whole body during exercise in the present study could also represent a factor to consider for the different in FENO drops. Indeed, given the involvement of the whole body during swimming compared to cycling, one could suggest at lower decrease in core temperature and consequently a lesser impact on bronchomotor nerves and NO synthase. Authors should discuss this hypothesis if this could represent a cofounding factor for this variable.

Thank you for that suggestion, we agree and have now included a sentence in the discussion (lines 299-306).

Round 2

Reviewer 1 Report

The manuscript has improved substantially although major concerns regarding sample size and design have not been addresses as this is not a possibility at this time. I do consider however that the study is now worthy of publication based on the changes provided by the authors. 

Reviewer 2 Report

thanks for providing corrections and clarifications

Reviewer 3 Report

I thank the authors for considering my request and respond to my comments. They completed the missing points and responded accordingly to my main comments. The current version of the manuscript is better written and could now be considered for publication.